# SA-MLP: Distilling Graph Knowledge from GNNs into Structure-Aware MLP

**Jie Chen**  *chenj19@fudan.edu.cn*
*Fudan University*

**Mingyuan Bai**  *mingyuan.bai@riken.jp*
*RIKEN AIP*

**Shouzhen Chen**  *chensz19@fudan.edu.cn*
*Fudan University*

**Junbin Gao**  *junbin.gao@sydney.edu.au*
*University of Sydney*

**Junping Zhang**  *jpzhang@fudan.edu.cn*
*Fudan University*

**Jian Pu**  *jianpu@fudan.edu.cn*
*Fudan University*

**Reviewed on OpenReview:** *https://openreview.net/forum?id=MZ2kKZc8m7*

## Abstract

The recursive node fetching and aggregation in message-passing cause inference latency when deploying Graph Neural Networks (GNNs) to large-scale graphs. One promising inference acceleration direction is to distill GNNs into message-passing-free student Multi-Layer Perceptrons (MLPs). However, the MLP student without graph dependency cannot fully learn the structure knowledge from GNNs, which causes inferior performance in heterophilic and online scenarios. To address this problem, we first design a simple yet effective Structure-Aware MLP (SA-MLP) as a student model. It utilizes linear layers as encoders and decoders to capture features and graph structures without message-passing among nodes. Furthermore, we introduce a novel structure-mixing knowledge distillation technique. It generates virtual samples imbued with a hybrid of structure knowledge from teacher GNNs, thereby enhancing the learning ability of MLPs for structure information. Extensive experiments on eight benchmark datasets under both transductive and online settings show that our SA-MLP can consistently achieve similar or even better results than teacher GNNs while maintaining as fast inference speed as MLPs. Our findings reveal that SA-MLP efficiently assimilates graph knowledge through distillation from GNNs in an end-to-end manner, eliminating the need for complex model architectures and preprocessing of features/structures. Our code is available at `https://github.com/JC-202/SA-MLP`.

## 1 Introduction

Graph Neural Networks (GNNs) (Kipf & Welling, 2017; Hamilton et al., 2017) have rapidly gained prominence for analyzing graph datasets in diverse domains such as social networks (Sankar et al., 2021), traffic networks (Wang et al., 2020) and recommendation systems (He et al., 2020; Liu et al., 2021). The primary success of GNNs stems from the message-passing mechanism (Gilmer et al., 2017) that extracts graph knowledge by aggregating neighborhood features via graph structures. This iterative aggregation process help nodes capture long-range neighbor and structure information, ultimately generating a more expressive node representation for downstream tasks. However, the number

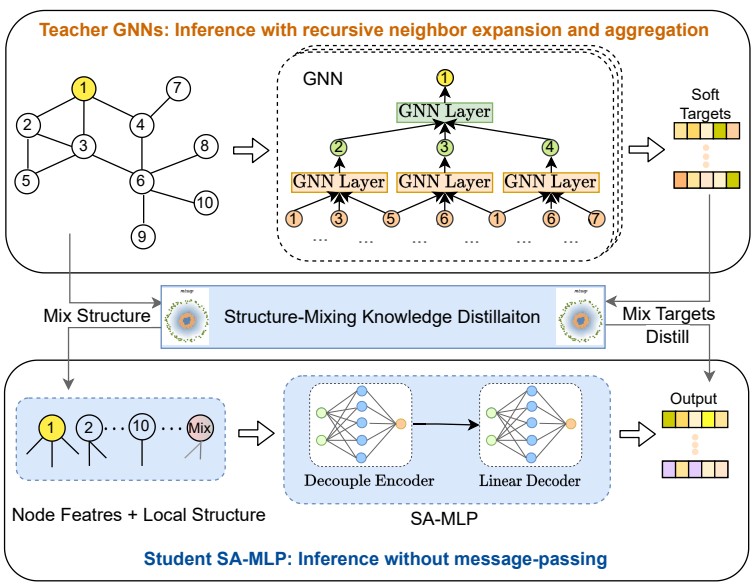

Figure 1: An overview of our distillation framework. A structure-aware MLP student learns from GNNs via a structure-mixing KD strategy to achieve substantially faster inference without sacrificing performance.

of neighbors for each node would exponentially increase as the number of layers increases (Zhang et al., 2022; Yan et al., 2020). Therefore, the recursive neighbor fetching induced by message-passing leads to inference latency, making GNNs hard to deploy for latency-constrained applications, especially for large-scale graphs (Hu et al., 2020).

Common inference acceleration methods, such as pruning (Zhou et al., 2021) and quantization (Tailor et al., 2021; Zhao et al., 2020), can speed up GNNs to some extent by reducing the Multiplication-and-ACcumulation (MAC) operations. However, they are limited by the recursive aggregation of GNNs. Knowledge distillation (KD) is another generic neural network learning paradigm for deployment that transfers knowledge from high-performance but resource-intensive teacher models to resource-efficient students (Hinton et al., 2015). Motivated by the impressive results of MLP-like models in computer vision (Melas-Kyriazi, 2021; Liu et al., 2022), one promising direction is to utilize KD to learn an efficient student MLP with the knowledge from pre-trained teacher GNNs (Zhang et al., 2022; Tian et al., 2023). After mimicking the output (soft labels) of GNNs, student MLPs can be deployed for faster inference while maintaining performance.

However, limited by the model structure and distillation strategy, MLPs are still unable to fully learn graph knowledge from GNNs. Specifically, 1) The standard MLPs only process node features (Zhang et al., 2022) or pre-computed DeepWalk embedding (Tian et al., 2023). The performance gain when distilling GNNs into MLPs can be primarily attributed to the strong memory capabilities (Szegedy et al., 2013) of MLPs, i.e., large MLPs can memorize outputs of GNNs on all observed nodes under transductive scenarios (Zhang et al., 2022). Therefore, the distilled MLP may fail in heterophilic scenarios, where node labels are highly correlated with structural information (Zhu et al., 2020). Similarly, in online scenarios (Si et al., 2023) where test nodes are not available to be pre-computed and distilled during training, the MLP may also struggle. 2) Compared to other complex relational or hidden distillation methods, logit-based KD benefits from marginal computational and storage costs (Zhao et al., 2022). Although logit-based KD is easy to deploy in practical applications, its performance is inferior when distilling GNNs into MLPs (Zhang et al., 2022). The reason lies in the logits' limited capacity to convey comprehensive graph knowledge. Hence, the supervision signal of logit-based KD is too sparse to transfer the graph knowledge from GNNs to MLPs effectively.

To address the above problems, as shown in Figure 1, we first design a simple yet effective Structure-Aware MLP (SA-MLP) student model, which can process structure inputs in an end-to-end and low-latency manner. It utilizes two linear encoders to explicitly encode node features and local structure information separately. Then, it concatenates them for another linear decoder to generate the final prediction. The model consists of only two layers, naturally incorporating structure information while offering faster inference. Second, we propose a novel structure-mixing knowledge distillation via the *mixup* (Zhang et al., 2018) technique to enhance the distillation density and improve the

structure-awareness of student MLP. It generates the virtual samples, which mix structures and soft labels of two nodes as additional supervision signals to guide the student MLP learning graph knowledge. This distillation strategy still only requires logits of GNNs, bypassing internal details of GNNs, making it lightweight and easy to use. Remarkably, we demonstrate that even without a sophisticated design, the linear layers of the SA-MLP and the logits from GNNs are sufficient for transferring graph knowledge to the student MLPs.

We conduct extensive experiments on eight public benchmark datasets under *transductive* and *online* scenarios (Si et al., 2023). The results demonstrate that SA-MLP surpasses other MLP-like students and attains comparable or even superior performance to teacher GNNs across all scenarios while offering significantly faster inference. Our contributions are summarized as follows:

- We propose a message-passing free model SA-MLP, which incorporates structure information without interaction among nodes and maintains low latency inference.

- We design a structure-mixing knowledge distillation strategy that improves the performance and structure awareness of SA-MLP.

- Extensive experimental results demonstrate that SA-MLP can achieve competitive performance as GNNs while inferring as fast as MLPs in both *transductive* and *online* scenarios.

## 2 Preliminary

### 2.1 Notation and Problem Setting

Consider a graph $\mathcal{G} = (\mathcal{V}, \mathcal{E})$, with $N$ nodes and $E$ edges. Let $\mathbf{A} \in \mathbb{R}^{N \times N}$ be the adjacency matrix, with $\mathbf{A}_{i,j} = 1$ if edge$(i, j) \in \mathcal{E}$, and 0 otherwise. Let $\mathbf{D} \in \mathbb{R}^{N \times N}$ be the diagonal degree matrix. Each node $v_i$ is given a $d$-dimensional feature representation $\mathbf{x}_i$ and a $c$-dimensional one-hot class label $\mathbf{y}_i$. The feature inputs are then formed by $\mathbf{X} \in \mathbb{R}^{N \times d}$, and the labels are represented by $\mathbf{Y} \in \mathbb{R}^{N \times c}$. The labeled and unlabeled node sets are denoted as $\mathcal{V}_L$ and $\mathcal{V}_U$, and we have $\mathcal{V} = \mathcal{V}_L \cup \mathcal{V}_U$. The task of node classification is to predict the labels $\mathbf{Y}$ by exploiting the nodes' features $\mathbf{X}$ and the graph structure $\mathbf{A}$.

### 2.2 Graph Neural Networks

Most existing GNNs follow the message-passing paradigm which contains node feature transformation and information aggregation from connected neighbors on the graph structure Gilmer et al. (2017); Wu et al. (2021). The general $k$-th layer graph convolution for a node $v_i$ can be formulated as

$$\mathbf{h}_i^{(k)} = f\left(\mathbf{h}_i^{(k-1)}, \left\{\mathbf{h}_j^{(k-1)} : j \in \mathcal{N}(v_i)\right\}\right), \tag{1}$$

where representation $\mathbf{h}_i$ is updated iteratively in each layer by collecting messages from its neighbors denoted as $\mathcal{N}(v_i)$. The graph convolution operator $f$ is usually implemented as a weighted sum of node representations according to the adjacent matrix $\mathbf{A}$ as in GCN Kipf & Welling (2017) and GraphSAGE Hamilton et al. (2017) or the attention mechanism in GAT Veličković et al. (2018). However, this recursive expansion and aggregation of neighbors cause inference latency, because the number of neighbors fetching will exponentially increase with the increasing number of layers Zhang et al. (2022); Yan et al. (2020).

The objective function for training GNNs is the cross-entropy of the ground truth labels $\mathbf{Y}_L$ and the output of the network $\hat{\mathbf{Y}}_L$ in labeled sets $\mathcal{V}_L$:

$$\mathcal{L}_{CE}(\hat{\mathbf{Y}}_L, \mathbf{Y}_L) = -\sum_{i \in \mathcal{V}_L} \sum_{j=1}^{c} \mathbf{Y}_{ij} \ln \hat{\mathbf{Y}}_{ij}. \tag{2}$$

### 2.3 Distill GNNs into MLPs

Knowledge distillation Hinton et al. (2015) aims to compress knowledge in a pre-trained large teacher model into a compact and fast-to-execute student model. The key idea is to force small student networks to imitate the soft targets

generated by the teachers, e.g., minimize the Kullback–Leibler (KL) divergence between the logit of teacher and student. Specifically, GLNN proposes to learn an MLP student, i.e., $\mathbf{Y}^s = \text{MLP}(\mathbf{X})$ to mimic the output of teacher GNN, i.e., $\mathbf{Y}^t = \text{GNN}(\mathbf{X}, \mathbf{A})$ with the following distillation objective:

$$\mathcal{L} = (1 - \lambda)\mathcal{L}_{CE}(\mathbf{Y}^s_L, \mathbf{Y}_L) + \lambda\mathcal{L}_{KD}(\mathbf{Y}^s, \mathbf{Y}^t), \quad \mathcal{L}_{KD}(\mathbf{Y}^s, \mathbf{Y}^t) = \sum_{v \in \mathcal{V}} \text{KL}(\mathbf{y}^s_v, \mathbf{y}^t_v), \tag{3}$$

where $\lambda$ is a hyper-parameter controlling the strength of KD. The goal of our paper is to utilize soft labels of GNNs to learn MLP-like students that can incorporate both features and structure information. Such that the learned MLP can deploy to both transductive and online scenarios, with a much lower computational cost while achieving similar or even better performance compared with GNNs.

## 3 Related Work

### 3.1 GNNs and Inference Acceleration

Most GNNs follow the message-passing mechanism Gilmer et al. (2017). For example, GCN Kipf & Welling (2017) aggregates local neighbor information according to the Laplacian matrix, GAT Veličković et al. (2018) employs attention in the aggregation, GraphSAGE Hamilton et al. (2017) introduces learnable aggregator functions to incorporate the local neighborhood, and GCNII Chen et al. (2020) introduces residual and initial connections. However, all of these methods suffer from inference latency induced by recursive aggregation. Some existing work focuses on speeding up GNN inference from the model compression perspective by pruning GNN parameters Zhou et al. (2021) and quantizing with low-precision integer arithmetic Zhao et al. (2020), such as Binarized DGCNN Bahri et al. (2021) and Degree-quant Tailor et al. (2021). Although these approaches can reduce model parameters and MAC operations, they are still limited by the neighbor-fetching latency. By using contrastive learning to train an MLP, Graph-MLP also attempts to avoid neighbor fetching Hu et al. (2021), but it only considers transductive rather than more practical online settings. There are also some works for neighbor sampling Zou et al. (2019); Chen et al. (2018), normalization Kose & Shen (2023), and sparsification Kose & Shen (2023) to speed up GNN training, which are complementary to our goal of inference acceleration.

### 3.2 Knowledge Distillation for GNNs

Existing GNN KD models try to distill large GNNs into smaller GNNs Zheng et al. (2022); Joshi et al. (2021; 2022). For instance, LSP Yang et al. (2020) and TinyGNN Yan et al. (2020) conduct KD while preserving local information, whereas GFKD Deng & Zhang (2021) and DFAD Zhuang et al. (2022) achieve graph-level KD via graph generation and adversarial training. Moreover, CPF Yang et al. (2021) utilizes KD to learn a label propagation student and enjoy prior knowledge. However, these message-passing-based methods still require information-fetching, resulting in inference latency Zhang et al. (2022). To eliminate message-passing, GLNN Zhang et al. (2022) is proposed that a teacher GNN teaches pure MLP student graph knowledge via KD. However, the MLP student may fail when structure information is essential to classification. It implies that the MLP does not fully learn structure information. NOSMOG Tian et al. (2023) incorporates positional features from DeepWalk Perozzi et al. (2014) into an MLP and utilizes adversarial learning with relational distillation to transfer graph knowledge from the hidden embeddings of GNNs. An alternative method involves using VQ-VAE and GNN's hidden information for pretraining a structure codebook Yang et al. (2024); Luo et al. (2024). However, these approaches require knowledge of the GNN's architecture and hidden information for distillation. Moreover, the pre-computation of DeepWalk embeddings or pretraining codebooks on large graphs can be costly. In contrast, our SA-MLP model doesn't require any pre-computation. It can learn structural knowledge end-to-end using a structure-mixing distillation strategy, which only requires soft labels of GNNs.

## 4 Structure-Aware MLP

### 4.1 Student MLP Model

The required information for node classification contains two parts, i.e., node feature and graph structure Kipf & Welling (2017); Zhu et al. (2021), which can utilize by GNNs. However, the current MLP-like student ignores the

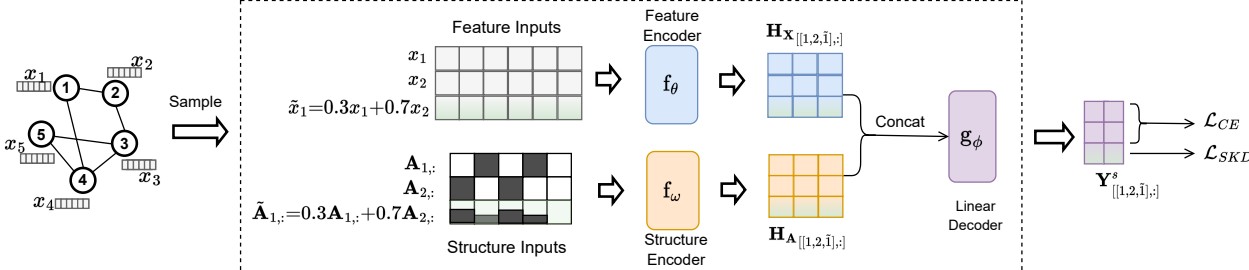

Figure 2: Illustration of SA-MLP, which encodes the feature and structure information with a decouple encoder and utilizes a linear decoder to generate final predictions. Moreover, to improve the learning ability of structure knowledge from teacher GNNs, we utilize structure-mixup to generate a virtual sample $\tilde{v}_1$ that contains hybrid structure knowledge to enhance the distillation density.

structure inputs for faster inference failed to thoroughly learn structure knowledge from teacher GNNs. To effectively exploit features and structure information to generate the final prediction, as shown in Figure 2, we propose the simple yet effective Structure-Aware MLP (SA-MLP) model with the corresponding decouple (feature/structure) encoder and a decoder. For inference efficiency, all these modules are implemented with *one* linear layer.

### 4.1.1 Decouple Encoder

For the feature encoder, same as GNNs, we simply utilize a linear layer to transform input features $\mathbf{X}$ into feature embedding that contains self-information. However, unlike GNNs, to design an efficient structure encoder, the key question is how to capture the structure information in $\mathbf{A}$ without interaction among nodes. Inspired by a natural language processing model FastText Joulin et al. (2016) which efficiently represents the sentence (line-structured of sequential words) as the summation of a bag of word embeddings, we treat each node's local structure, i.e., $\mathbf{A}_{i,:}$, as a bag of context nodes for $v_i$ to extract structure information. This idea can naturally be implemented by considering columns of $\mathbf{A}$ as features and feeding them into a linear layer to capture structure information. Therefore, we encode feature and structure information individually to get feature embedding $\mathbf{H_X} \in \mathbb{R}^{n \times d_1}$ and structure embedding $\mathbf{H_A} \in \mathbb{R}^{n \times d_2}$ by

$$\mathbf{H_X} = f_\theta(\mathbf{X}) = \mathbf{X}\mathbf{W_X} + \mathbf{b_X}, \tag{4}$$

$$\mathbf{H_A} = f_\omega(\mathbf{A}) = \mathbf{A}\mathbf{W_A} + \mathbf{D_A}, \tag{5}$$

where mapping $\mathbf{W_X} \in \mathbb{R}^{d \times d_1}$, bias $\mathbf{b_X} \in \mathbb{R}^{1 \times d_1}$, and mapping $\mathbf{W_A} \in \mathbb{R}^{N \times d_2}$ are learnable parameters. Note that, in contrast to single-layer GNNs, the mapping $\mathbf{W_A}$ for $\mathbf{A}$ serves as a learnable positional embedding table for each node, akin to a word embedding table in natural language processing. This approach offers better generalization capabilities compared to the pre-computed DeepWalk positional embeddings utilized in NOSMOG Perozzi et al. (2014), Tian et al. (2023). Moreover, we inject node centrality in the bias $\mathbf{D_A} \in \mathbb{R}^{N \times d_2}$, which measures how important a node is in the graph and is usually a strong signal for graph understanding Ying et al. (2021). Specifically, the degree bias $\mathbf{D_A}_{i,:}$ of $v_i$ is learnable embedding vector specified by its degree.

### 4.1.2 Linear Decoder

To generate the final prediction, we fuse feature and structure information via concatenation and utilize another one layer linear decoder $g_\phi$:

$$\mathbf{Y}^s = g_\phi(\sigma[\mathbf{H_X} || \mathbf{H_A}]), \tag{6}$$

where $\sigma$ indicates a ReLU activation function. Here, we also tested other fusion mechanisms and increased the number of layers for $g_\phi$ but did not observe noticeable improvement, see Table 5. The reason a linear decoder suffices might be that the decoupled encoder has already learned rich graph knowledge through distillation, hence a simple decoder exhibits good generalization capabilities.

### 4.1.3  Scalability and Complexity

Compared to GNNs, SA-MLP discards the recursive aggregation and simplifies the handling of structural information by treating the columns of $\mathbf{A}$ as input features for $f_\omega$. This approach allows SA-MLP to maintain the speed of standard MLPs, benefiting from the sparsity of real-world adjacency matrices $\mathbf{A}$ and the efficient sparse-dense matrix multiplication with $\mathbf{W_A}$. Moreover, although $N$ may be large, such $\mathbf{W_A}$ is similar to the widely used user/item embedding in large-scale industrial recommendation systems He et al. (2017); Barkan & Koenigstein (2016), which is scalable due to the embedding infrastructure. When the system is dynamically increasing in the online scenarios, it possesses scalability by collecting new nodes/edges and appends node ID embedding into $\mathbf{W_A}$ for periodic retraining.

The time complexity of SA-MLP in each full-batch forward pass is $O(dE + Nd^2K)$, in which $d$ is the hidden dimension, $N$ is the number of nodes, $E$ is the number of edges, and $K$ is the number of layers. The cost is $O(dE)$ for the first linear mapping of $\mathbf{A}$ and $O(d^2)$ for each MLP. In fact, the operations of mapping $\mathbf{A}$ can be easily implemented in the sparse matrix form, which results in high time efficiency. While most GNNs have to propagate features using the adjacency in each layer, their complexity is usually $O(dKE + Nd^2K)$, and the $O(dKE)$ term makes it difficult to deploy to large-scale graphs Yan et al. (2020).

In terms of memory complexity, SA-MLP requires additional storage for the structural embeddings $\mathbf{W_A}$, leading to a memory footprint of $O(Nd)$. However, during the stages of model training and inference, $\mathbf{W_A}$ is conveniently stored as an embedding table within parameter servers Li et al. (2013), thus only the pertinent parameters required for each individual batch are retrieved. According to the structure of each batch's lines within the adjacency matrix A, the model extracts the structural embeddings of $D$ neighbors for $B$ nodes from $\mathbf{W_A}$, giving rise to a memory complexity of $O(BDd)$. This efficient design ensures manageable memory usage even under vast scales.

### 4.2  Structure-Mixing Knowledge Distillation

Although SA-MLP can efficiently encode features and structure information, discarding the message-passing among nodes still causes suboptimal performance. To allow SA-MLP to enjoy both the efficiency of MLP and the accuracy of GNNs, we conduct cross-model KD from GNNs to SA-MLP. Since the outputs of GNNs are considered to include structure information Yang et al. (2021); Yan et al. (2020), the logit-based KD Hinton et al. (2015) has been utilized to extract graph knowledge from GNNs.

Inspired by the *mixup* data augmentation strategy in computer vision Zhang et al. (2018), and without relying on sophisticated KD designs and additional information about GNNs, we propose a structure-mixup variant to enhance the density of graph knowledge distillation. *Mixup* generates virtual samples via a linear combination of paired inputs $\mathbf{X}$ and labels $\mathbf{Y}$. It can enhance the generalization of models by increasing the density of the vicinal data distribution for training Zhang et al. (2018). Our proposed *structure-mixup* simultaneously mixes features $\mathbf{X}$, structure $\mathbf{A}$, and teacher's output $\mathbf{Y}^t$ to generate virtual distillation samples as follows.

$$\text{Structure-Mixup} \begin{cases} \gamma \sim Beta(\alpha, \alpha) \\ \tilde{\mathbf{X}} = \gamma \mathbf{X} + (1-\gamma)\mathbf{X}_{*,:} \\ \tilde{\mathbf{A}} = \gamma \mathbf{A} + (1-\gamma)\mathbf{A}_{*,:} \\ \tilde{\mathbf{Y}}^t = \gamma \mathbf{Y}^t + (1-\gamma)\mathbf{Y}^t_{*,:} \end{cases} \tag{7}$$

where the hyper-parameter $\alpha$ controls the strength of interpolation. The subscript $_*$ means the index of the corresponding batch sample pair after random shuffling for linear combination, e.g., the row index of nodes from [1,2,...,n] to [5,n-1,...,2] after shuffling. The mixed pair of structure $\tilde{\mathbf{A}}$, features $\tilde{\mathbf{X}}$, and the teacher's output $\tilde{\mathbf{Y}}^t$ contains a more comprehensive set of relations between structure inputs and labels, enhancing the density of distillation from the teacher GNN to the SA-MLP.

Note that the *mixup* on graphs is challenging due to the irregularity and connectivity of the input structure $\mathbf{A}$ for GNNs. Existing *mixup* methods for GNNs attempt to blend hidden embeddings through complex structure alignment techniques Wang et al. (2021); Han et al. (2022). However, our approach circumvents the challenge about how to adapt the mixed structure $\mathbf{A}$ to GNNs. Instead, we focus on the mixed outputs $\tilde{\mathbf{Y}}^t$ from existing teacher GNNs, which serve as additional supervision signals to transfer knowledge to student MLPs. Then, the mixed structure $\tilde{\mathbf{A}}$ and features $\tilde{\mathbf{X}}$

can be naturally processed by SA-MLP to generated student output $\tilde{\mathbf{Y}}^s$.

$$\tilde{\mathbf{Y}}^s = \text{SA-MLP}(\tilde{\mathbf{X}}, \tilde{\mathbf{A}}). \tag{8}$$

To the best of our knowledge, we are the first to introduce the *mixup* in the KD of GNNs to enhance the distillation density. To transfer the graph knowledge from GNNs to MLP, we minimize the Kullback–Leibler (KL) divergence distance between the hybrid output $\tilde{\mathbf{Y}}^s$ of students and $\tilde{\mathbf{Y}}^t$ of teachers. Compared with relational or hidden distillation, we bypasses the detail of GNNs and still only requires the soft outputs of teachers, therefore benefits from marginal computational and storage costs. The overall objective on this structure-mixing knowledge distillation (SKD) is:

$$\mathcal{L}_{SKD}(\mathbf{Y}^s, \mathbf{Y}^t) = \sum_{v \in \tilde{\mathcal{V}}} \text{KL}(\tilde{\mathbf{y}}_v^s, \tilde{\mathbf{y}}_v^t). \tag{9}$$

### 4.3 Overall Training and Inference

In this section, we describe the overall training and inference process of transductive and online settings for SA-MLP.

#### 4.3.1 Transductive:

In the transductive setting, the model can observe the structure and features of all nodes during training. Hence, the training objective contains the cross-entropy loss with the ground-truth label on training nodes and the distillation loss with the output of teacher GNNs on total nodes. The total objective is:

$$\mathcal{L} = (1 - \lambda)\mathcal{L}_{CE}(\mathbf{Y}_L^s, \mathbf{Y}_L) + \lambda \mathcal{L}_{SKD}(\mathbf{Y}^s, \mathbf{Y}^t). \tag{10}$$

However, the transductive setting may not be sufficient to evaluate the graph knowledge learning ability since MLP-like models may memorize all the outputs of teacher GNNs.

#### 4.3.2 Online:

In the online setting, the model can only observe the structure and features of training set nodes. Hence the total objective only involves the training nodes:

$$\mathcal{L} = (1 - \lambda)\mathcal{L}_{CE}(\mathbf{Y}_L^s, \mathbf{Y}_L) + \lambda \mathcal{L}_{SKD}(\mathbf{Y}_L^s, \mathbf{Y}_L^t). \tag{11}$$

When inference, the SA-MLP can utilize the connection from training nodes to the newest nodes, which contain learned structure information.

## 5 Experiments

### 5.1 Experimental Setup

#### 5.1.1 Datasets

To evaluate the performance of the proposed SA-MLP, we consider eight public benchmark datasets, including three citation datasets Sen et al. (2008) (Cora, Citeseer, Pubmed), two larger OGB datasets Hu et al. (2020) (Arxiv, Products), and three heterophily datasets (Chameleon, Squirrel, Arxiv-year) Pei et al. (2020); Lim et al. (2021) whose structure information is important. We used the standard public splits of OGB datasets, and ten frequently used fully supervised splits (48%/32%/20% of nodes per class for train/validation/test) provided by Pei et al. (2020); Zhu et al. (2020) of other datasets for a fair comparison and reproduction. Note that for three small citation datasets, these splits reduce randomness and the possibility of overfitting Zhu et al. (2020); Pei et al. (2020), which are stricter than the random splits used in GLNN Zhang et al. (2022). The statistics are shown in Table 1 and more details can be found in the Appendix.

Table 1: Statistics of the datasets

| Datasets | Types | #Nodes | #Edges | #Features | #Classes |
|---|---|---|---|---|---|
| Cora | Homophily | 2,708 | 5,429 | 1,433 | 7 |
| Citeseer | Homophily | 3,327 | 4,732 | 3,703 | 6 |
| Pubmed | Homophily | 19,717 | 44,324 | 500 | 3 |
| Arxiv | Homophily | 169,343 | 1,166,243 | 128 | 40 |
| Products | Homophily | 2,449,029 | 61,859,140 | 100 | 47 |
| Chameleon | Heterophily | 2,277 | 36,101 | 2,325 | 5 |
| Squirrel | Heterophily | 5,201 | 217,073 | 2,089 | 5 |
| Arxiv-year | Heterophily | 169,343 | 1,166,243 | 128 | 5 |

### 5.1.2  Transductive and Online Setting

For the transductive setting, we use all node features and structures for training and distillation. For the online setting, we hold out all test and validation nodes ($\mathcal{V}_U$) with their connections when training. For every split, we extract graph $\mathcal{G}$ to the subgraph $\mathcal{G}_L$ that only contains nodes $\mathcal{V}_L$ with corresponding edges, and the subgraph $\mathcal{G}_{online}$ including $\mathcal{G}_L$ plus such edges from $\mathcal{V}_L$ to $\mathcal{V}_U$.

- Transductive: train on $(\mathcal{G}, \mathbf{X}, \mathbf{Y}_L)$, KD for all nodes $\mathcal{V}$, evaluate on $(\mathcal{G}, \mathbf{X}_U, \mathbf{Y}_U)$.

- Online: train on $(\mathcal{G}_L, \mathbf{X}_L, \mathbf{Y}_L)$, KD for $\mathcal{V}_L$; evaluate on $(\mathcal{G}_{online}, \mathbf{X}_U, \mathbf{Y}_U)$.

### 5.1.3  Baselines and Training Details

In the following experiments, as in Zhang et al. (2022), we also use GraphSAGE Hamilton et al. (2017) as our basic teacher model to investigate the learning ability of the proposed SA-MLP from GNNs. Moreover, for the heterophily datasets, we apply residual connections to improve the performance of GraphSAGE. Following the standard setting Hu et al. (2020); Bo et al. (2021), we fix the hidden dimension of SA-MLP as 128 for all datasets except 64 for Products. The number of layer in SA-MLP is 2 for all datasets for inference efficiency. We use Adam Kingma & Ba (2014) for optimization, LayerNorm Ba et al. (2016), and tune other hyper-parameters, including dropout rate from [0, 0.2, 0.5], learning rate from [0.01, 0.005, 0.05], weight decay from [0, 5e-4, 5e-5], $\delta$ from [0, 0.2, 0.5], and $\lambda$ from [0.5, 0.8, 1] for distillation via validation sets of each dataset. We report results of GLNN and NOSMOG with the same experimental setup if available. If the results were not previously reported, we conducted a hyper-parameter search based on the official codes. More comparisons with NOSMOG and GLNN under random splits can be found in the Appendix.

## 5.2  Overall Performance

Experimental results on eight datasets over two scenarios with teacher GNN, MLP, state-of-the-art GNN-MLP distillation methods GLNN and NOSMOG, and SA-MLP are presented in Table 2 and 3. In summary, due to the structure awareness, our SA-MLP consistently achieves the best performance and we further observe the following results:

### 5.2.1  Transductive

As shown in Table 2, SA-MLP outperforms all baselines, including teacher GNN models across all datasets. The reason is that SA-MLP encodes the structure information in an alternative way, which may be complementary to GNNs, especially in the heterophily datasets (Chameleon, Squirrel and Arxiv-year). Comparing SA-MLP to NOS-MOG, our SA-MLP achieves an improvement of up to $3.88\%$. Note that SA-MLP is end-to-end learning without the pre-computed deep-walk positional embedding, the relational distillation and adversarial learning used in NOSMOG, demonstrating the simplicity but effectiveness of SA-MLP.

### 5.2.2 Online

As shown in Table 3, we have the following observations: 1) GLNN only improves slightly on MLP and retains a large performance gap compared with teacher GNNs. GLNN without graph dependency failed to mimic the output of GNNs in test nodes. Because distillation only occurs on the training nodes. 2) NOSMOG performs better than GLNN by incorporating the pre-computed deep-walk positional embedding and the relational distillation with adversarial learning. However, in most cases, it is still inferior to GNNs due to the limited generalization of deep-walk in the test nodes in online settings. 3) SA-MLP still consistently outperforms the others across all datasets since the SA-MLP can utilize the connection from training nodes to the newest test nodes. Combined with structure-mixing distillation, such structure information improve the generalization ability of SA-MLP in online settings. Hence the performance of SA-MLP is even beyond teacher GNN.

Table 2: Experiment results in node classification for the *transductive* setting: We report the mean test accuracy (%) and standard deviation over ten runs. $\triangle$ represents the improvement of SA-MLP, i.e., $\triangle_{GNN} \geq 0$ indicates that it outperforms GNN.

| Dataset | SAGE | MLP | GLNN | NOSMOG | SA-MLP | $\triangle_{GNN}$ | $\triangle_{MLP}$ | $\triangle_{NOSMOG}$ |
|---|---|---|---|---|---|---|---|---|
| Cora | 86.14±0.74 | 74.75±2.22 | 86.21±1.42 | 87.16±1.36 | **87.26** ±0.89 | 1.12(1.28%) | 12.51(14.51%) | 0.10(0.12%) |
| Citeseer | 75.13±2.28 | 72.41±2.18 | 76.15±2.19 | 76.34±2.14 | **76.67** ±1.17 | 1.54(2.02%) | 4.26(5.59%) | 0.33(0.44%) |
| Pubmed | 89.17±0.46 | 86.65±0.35 | 89.32±0.43 | 89.69±0.37 | **90.31** ±0.38 | 1.14(1.27%) | 3.66(4.10%) | 0.62(0.70%) |
| Arxiv | 70.73±0.35 | 56.05±0.46 | 63.46±0.45 | 71.64±0.29 | **71.68** ±0.18 | 0.95(1.33%) | 15.63(24.63%) | 0.04(0.06%) |
| Product | 77.17±0.32 | 62.47±0.10 | 68.86±0.46 | 78.45±0.38 | **79.02** ±0.15 | 1.85(2.36%) | 16.55(24.03%) | 0.57(0.74%) |
| Chameleon | 70.61±1.76 | 46.36±2.52 | 67.98±1.71 | 70.72±2.88 | **71.09** ±2.21 | 0.48(0.68%) | 24.73(36.38%) | 0.37(0.52%) |
| Squirrel | 62.51±2.01 | 33.18±1.94 | 62.23±1.87 | 62.88±1.83 | **63.51** ±2.05 | 1.00(1.59%) | 30.33(48.74%) | 0.63(1.01%) |
| Arxiv-year | 51.85±0.22 | 36.71±0.21 | 46.22±0.20 | 52.34±0.18 | **54.35** ±0.15 | 2.50(4.78%) | 17.64(38.17%) | 2.01(3.88%) |

Table 3: Experiment results in node classification for the *online* setting.

| Dataset | SAGE | MLP | GLNN | NOSMOG | SA-MLP | $\triangle_{GNN}$ | $\triangle_{MLP}$ | $\triangle_{NOSMOG}$ |
|---|---|---|---|---|---|---|---|---|
| Cora | 80.78±2.44 | 74.75±2.22 | 74.98±1.84 | 77.52±2.86 | **83.60** ±1.78 | 2.82(3.64%) | 8.85(11.80%) | 6.08(7.53%) |
| Citeseer | 73.24±1.73 | 72.41±2.18 | 72.55±1.79 | 73.42±1.59 | **75.15** ±1.91 | 1.91(2.60%) | 2.74(3.78%) | 1.73(2.36%) |
| Pubmed | 87.98±0.66 | 86.65±0.35 | 88.25±0.43 | 88.86±0.00 | **89.21** ±0.31 | 1.23(1.38%) | 2.56(2.90%) | 0.35(0.40%) |
| Arxiv | 67.69±0.24 | 56.05±0.46 | 56.35±0.21 | 63.66±0.25 | **68.01** ±0.24 | 0.32(0.50%) | 11.96(21.22%) | 4.35(6.43%) |
| Product | 65.55±0.88 | 62.47±0.10 | 62.45±0.34 | 65.32±0.42 | **67.46** ±0.36 | 1.91(2.92%) | 4.99(7.99%) | 2.14(3.26%) |
| Chameleon | 63.73±1.58 | 46.36±2.52 | 46.91±2.09 | 59.05±2.31 | **65.78** ±2.33 | 2.05(3.47%) | 19.42(41.40%) | 6.73(10.56%) |
| Squirrel | 55.65±1.47 | 33.18±1.94 | 33.27±1.78 | 44.16±2.24 | **56.96** ±1.80 | 1.31(2.97%) | 23.78(71.48%) | 12.80(23.00%) |
| Arxiv-year | 48.42±0.46 | 36.71±0.21 | 36.92±0.10 | 40.85±0.36 | **49.75** ±0.25 | 1.33(3.26%) | 13.04(35.32%) | 8.90(18.38%) |

### 5.3 Ablation Study

Table 4: Ablation studies of student MLPs and KDs, while the variants b and d remove the degree bias.

| | Transductive | | Online | |
|---|---|---|---|---|
| | Homo | Hete | Homo | Hete |
| GNNs | 83.48 | 66.94 | 80.67 | 59.69 |
| a)MLP(X)+KD | 83.89 | 65.11 | 78.59 | 38.55 |
| b)MLP(X,A)+KD | 84.59 | 66.68 | 82.29 | 58.94 |
| c)MLP(X,A,D)+KD | 84.62 | 67.01 | 82.51 | 59.01 |
| d)MLP(X,A)+SKD | 84.67 | 67.15 | 82.65 | 60.11 |
| e)MLP(X,A,D)+SKD | **84.75** | **67.30** | **82.65** | **61.37** |

Table 5: Ablation studies of different types of fusion and layers in decoder $g_\phi$ in student MLPs.

| | | Transductive | | Online | |
|---|---|---|---|---|---|
| | | Homo | Hete | Homo | Hete |
| Fusion | Add | 84.61 | 67.24 | 82.64 | 60.76 |
| | Attentive | **84.79** | 66.74 | 79.52 | 55.92 |
| Layer | 2 | 84.44 | 67.21 | 82.60 | 60.12 |
| | 3 | 84.56 | 67.13 | 82.53 | 60.05 |
| | Default | 84.75 | **67.30** | **82.65** | **61.37** |

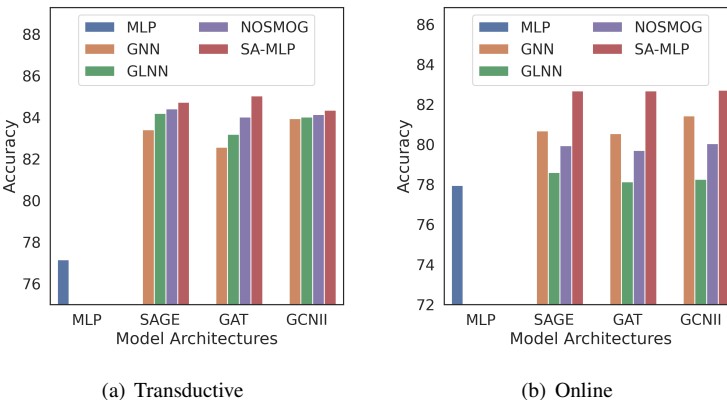

(a) Transductive  (b) Online

Figure 3: Mean accuracy over three citation datasets under different teacher architectures.

### 5.3.1 Variants of Student MLPs and KDs

We study the five variants of SA-MLP under homophily (Cora, Citeseer and Pubmed) and heterophily (Chameleon and Squirrel) in both transductive and online settings. Due to space limits, we report the average test accuracy of these variants, as shown in Table 4 and Table 5. Our findings reveal:

1) *Role of structure information in student MLP*: from the comparison among variants a, b, and c in Table 4, the structure information $\mathbf{A}$ is crucial for the performance of SA-MLP, especially in the online and heterophily scenarios. Moreover, the degree information $\mathbf{D}$ with centrality encoding in Equation equation 5 can further improve performance.

2) *Role of KD strategy*: By comparing variants b with d and c with e, we can find that the structure-mixing distillation strategy consistently outperforms the standard logit-based distillation for both $\mathrm{MLP}(\mathbf{X}, \mathbf{A})$ and $\mathrm{MLP}(\mathbf{X}, \mathbf{A}, \mathbf{D})$ student variants. And the improvement is more significant under the heterophilic graphs. The reason is that the node classification of these graphs is more structure-oriented, and the structure-mixing distillation can reduce the sparsity of structure knowledge to improve the structure awareness of student MLPs.

3) *Role of fusion and layers in decoder*: We study different fusion variants and layers of decoder $\mathrm{g}_\phi$ in SA-MLP as follows.

- *Add*: It replaces the concat of $\mathbf{H_X}$ and $\mathbf{H_A}$ to $\mathbf{H_X} + \mathbf{H_A}$.

- *Attentive*: It replaces the concat of $\mathbf{H_X}$ and $\mathbf{H_A}$ to $\gamma\mathbf{H_X} + (1-\gamma)\mathbf{H_A}$, where the attentive value $\gamma \in [0,1]^{N \times 1}$ of nodes is generated by another linear layer as $\mathrm{sigmoid}\,([\mathbf{H_A}||\mathbf{H_X}]\mathbf{W} + \mathbf{b})$.

- *Layer-2/3*: It replaces the linear layer in $\mathrm{g}_\phi$ with two or three layers of MLP.

The results in Table 5 show that these variants did not contribute to a noticeable improvement in various scenarios. The default fusion of concatenate of $\mathbf{H_X}$ and $\mathbf{H_A}$ and the one layer of $\mathrm{g}_\omega$ make SA-MLP simple but effective while maintaining fast inference.

### 5.3.2 Effects of Teacher GNN Architecture

We compare different teacher architectures, including SAGE Hamilton et al. (2017), GCNII Chen et al. (2020), and GAT Veličković et al. (2018). We choose three citation datasets in the *transductive* and *online* settings and report the mean accuracy. As shown in Figure 3, in the online settings, GLNN and NOSMOG suffer from weak generalization and cannot achieve similar results as teacher GNNs after being distilled. However, SA-MLP can achieve better results than different architecture teacher GNNs in all settings. The results show that structure awareness is the key to the model generalization ability, and our SA-MLP and distillation strategy are general and robust to accelerate the deployment of various GNNs.

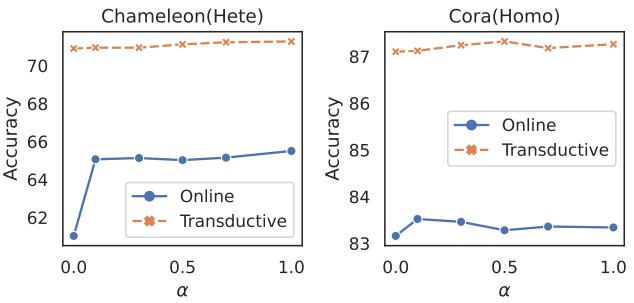 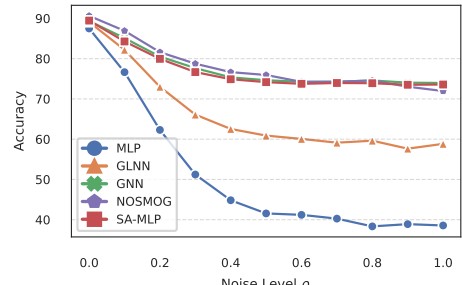

Figure 4: Sensitivity of parameter $\alpha$ in *mixup* of structure-mixing knowledge distillation.

Figure 5: Sensitivity analysis for varying levels of noise in node features.

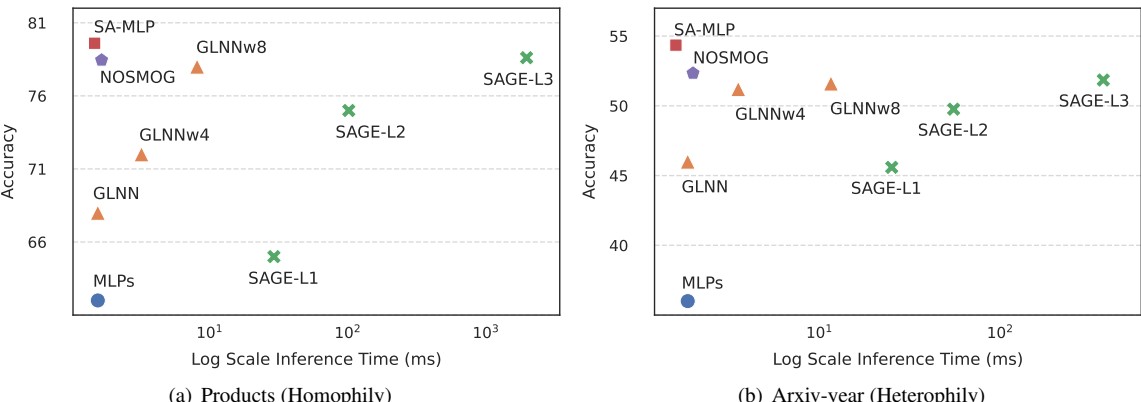

(a) Products (Homophily)

(b) Arxiv-year (Heterophily)

Figure 6: Accuracy vs. Inference Time among different layers of GNNs and different sizes of MLP models on large-scale datasets under transductive settings.

## 5.4 Sensitivity Analysis

### 5.4.1 Hyper-parameter Sensity of Structure-Mixing Knowledge Distillation

We also investigate the sensitivity of the parameter $\alpha$, which controls the strength of *mixup* in structure-mixing knowledge distillation. As shown in Figure 4, the $\alpha$ was effective in a wide range under both homophily and heterophily datasets in transductive and online scenarios. Moreover, it significantly increases the performance under the heterophilic dataset Chameleon in the Online scenario. The reason is that the structure information in heterophilic graphs and online scenarios is the key clue for generalization. Increasing the strength of structure-mixing can enhance the density of distillation to improve the structure knowledge of SA-MLP.

### 5.4.2 Robustness to Noisy Features

To illustrate the significance of structure information, following Zhang et al. (2022), we introduce varying levels of Gaussian noise to node features, denoted as $\mathbf{X} = (1-\rho)\mathbf{X} + \rho\epsilon$, where $\rho \in [0, 1]$ represents the noise level, and $\epsilon$ is the Gaussian noise independent of $\mathbf{X}$. We then visualize the mean results of the PubMed dataset in transductive and online settings at different noise levels $\rho$ in Figure 5. We observe that the performance of MLP and GLNN drop significantly as $\rho$ increases. However, SA-MLP remains comparable to teacher GNN and NOSMOG, as they both explicitly utilize structured inputs instead of relying solely on node features and NOSMOG employs adversarial training. Note that GNN conducts recursive aggregation from neighbors to extract graph knowledge and overcome noise, while SA-MLP directly processes structure inputs.

### 5.5 Efficiency Comparison

To further analyze the efficiency of SA-MLP, in Figure 6, we visualize the trade-off between prediction accuracy and inference time on large-scale Products and Arxiv-year datasets. Following GLNN Zhang et al. (2022), the log scale inference times (ms) are computed based on 10 randomly chosen nodes. We observe consistent trends in both homophily and heterophily, with SA-MLP providing an optimal balance of high accuracy and fast inference time. Notably, SA-MLP demonstrates performance improvement over MLP, GLNN and NOSMOG with similar inference speeds, especially for MLP and GLNN. While increasing the hidden size of GLNN to GLNNw4 (4 times wider than GLNN) and GLNNw8 (8 times wider than GLNN) improves their performance, they still lag behind SA-MLP and require longer inference times. In comparison to teacher GNNs that display similar performance to SA-MLP, these models demand considerably longer inference times, making them unsuitable for many large-scale data real-world applications. For instance, the 3-layer GraphSAGE (SAGE-L3) requires 1957ms on the Products dataset, whereas SA-MLP needs only 2ms, achieving a 978x inference speed up without sacrificing performance. Although NoSMOG also performs well, it requires preprocessing of the structure with DeepWalk, which is actually very slow on large graphs. However, this preprocessing time is not included in the inference time. In contrast, SAMLP can directly process the raw input without the need for additional preprocessing.

## 6   Conclusion

We have presented a message-passing free SA-MLP, a practical solution to address the deployment of GNNs via knowledge distillation. This is achieved by designing a simple yet effective structure-aware student MLP model and combining it with a novel structure-mixing knowledge distillation strategy. Experiments on eight benchmark datasets show that SA-MLP possesses both the accuracy of GNNs and the inference speed of MLPs, whether in homophilic or heterophilic graphs, as well as in transductive and online scenarios. One future work is to investigate the application of SA-MLP to other downstream tasks, such as graph classification and link prediction on social networks.

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

# A Supplementary Material of SA-MLP

## A.1 Details of Datasets

We provide the details of the five homophilic datasets (connected nodes tend to be the same label) and three heterophilic datasets (labels of connected nodes tend to be different) in the following:

- Homophilic Datasets

  - *Citeseer, Pubmed, Cora* Kipf & Welling (2017): For the basic citation datasets, nodes correspond to papers, edges correspond to citation links, the sparse bag-of-words are the feature representation of each node, and the label of each node represents the topic of the paper. Note that we use the public ten data split(48%/32%/20% for Train/Val/Test) in Pei et al. (2020); Zhu et al. (2020) to reduce randomness and enhance reproducibility. Compared with the GLNN that uses only 20 nodes of each class for training, the results of our splits are more stable and reduce the possibility of overfitting Zhu et al. (2020).
  - *Arxiv* Hu et al. (2020): The Arxiv dataset is a large-scale citation network collected from all Computer Science arXiv papers. Each node is an arXiv paper, and edges are citation relations between papers. The features are 128-dimensional averaged word embeddings of each paper, and labels are subject areas of papers.
  - *Products* Hu et al. (2020): The Products is a large-scale Amazon product co-purchasing network. Nodes represent products sold in Amazon, edges indicate the products purchased together, and features are 100-dimensional bag-of-words features.

- Heterophilic Datasets

  - *Squirrel, Chameleon* Pei et al. (2020): Chameleon and Squirrel are web pages extracted from different topics in Wikipedia. Similar to WebKB, nodes and edges denote the web pages and hyperlinks among them, respectively, and informative nouns in the web pages are employed to construct the node features in the bag-of-word form. Webpages are labeled in terms of the average monthly traffic level.
  - *Arxiv-year* Lim et al. (2021): Modifying node labels of the Arxiv dataset to the year of paper, and the goal is to predict the year of paper publication that allows for evaluation of GNNs in large-scale non-homophilous settings.

## A.2 Additional Comparison of Citation Datasets

We provide additional experiments for the comparison under the random splits (20 labeled nodes of each class during training) of citation datasets used in GLNN and report the mean test accuracy in Table 6. We can observe that SA-MLP also consistently outperforms GLNN, NOSMOG and teacher GNN. Moreover, we conduct the production (*prod*) scenario that involves both inductive (*ind*) and transductive (*trans*) settings used in GLNN (see Table 6). We find that SA-MLP can also achieve the best performance across all scenarios. However, the performance of these splits that only contained 20 labeled nodes is susceptible to hyper-parameters since the few training data cause the risk of overfitting.

Table 6: Inductive, transductive, and production scenario of citation datasets under random splits.

| Dataset | Eval | SAGE | GLNN | NOSMOG | SA-MLP |
|---------|------|------|------|--------|--------|
| Cora | *prod* | 79.53 | 77.82 | 81.02 | **81.21** |
| | *ind* | 81.03 | 73.21 | 81.36 | **81.03** |
| | *trans* | 79.16 | 78.97 | 80.93 | **81.47** |
| Citeseer | *prod* | 68.06 | 69.08 | 70.60 | **70.67** |
| | *ind* | 69.14 | 68.48 | 70.30 | **70.53** |
| | *trans* | 67.79 | 69.23 | 70.67 | **70.81** |
| Pubmed | *prod* | 74.77 | 74.67 | 75.82 | **76.12** |
| | *ind* | 75.07 | 74.52 | 75.87 | **76.04** |
| | *trans* | 74.70 | 74.70 | 75.80 | **76.15** |

Table 7: Comparison with GLNN+ in large-scale datasets.

| Dataset | Setting | SAGE | GLNN | GLNN+ | SA-MLP |
|---------|---------|------|------|-------|--------|
| Arxiv | *trans* | 70.92 | 63.46 | **72.15** | 71.54 |
| | *online* | 67.69 | 56.35 | 56.56 | **68.01** |
| Products | *trans* | 78.61 | 68.86 | 77.65 | **79.02** |
| | *online* | 65.55 | 62.45 | 62.58 | **67.46** |
| Arixv-year | *trans* | 51.85 | 46.22 | 51.02 | **53.31** |
| | *online* | 48.42 | 36.92 | 36.81 | **49.55** |

Table 8: Speed comparison between SA-MLP and other inference acceleration of SAGE. Numbers (in ms) are inference time on 10 randomly chosen nodes. "*" indicates our implementation based on released codes of GLNN.

| Model | Structure | Arxiv | Products |
|---|:---:|---|---|
| SAGE | ✓ | 489.49 | 2071.30 |
| QSAGE | ✓ | 433.90 | 1946.49 |
| PSAGE | ✓ | 465.43 | 2001.46 |
| NSSAGE | ✓ | 91.03 | 107.31 |
| GLNN+ | | 3.34 | 7.56 |
| SAGE* | ✓ | 386.85 | 1957.11 |
| GLNN+* | | 3.45 | 8.64 |
| NOSMOG | ✓ | 1.36 | 1.34 |
| SA-MLP | ✓ | **1.18** | **1.12** |

## A.3 Additional Comparison of GLNN+

GLNN also provides a larger GLNN+, which scales hidden dimension from 256 to 1024 for Arxiv (GLNNw4) and 2048 for Product (GLNNw8), with a larger capacity but a slower speed. We provide additional experiments for the GLNN+ of the *trans* and *online* for large-scale OGB datasets. We omit other datasets since the performance of GLNN+ is similar to that of GLNN. From Table 7, we find that GLNN+ can improve the performance of large-scale datasets under the *trans* setting. However, it achieves similar results to GLNN under the *online* setting, which implies that the improvement of *trans* for OGB datasets is due to the memory capacity, i.e., the larger parameters of GLNN+ can memorize all the teacher outputs. It still does not fully understand the structure information and generalizes limitedly on unseen test nodes under the *online* setting. However, the improvement over both *trans* and *online* of our SA-MLP is due to explicit structure awareness.

## A.4 Additional Inference Time Comparison

Following the settings GLNN Zhang et al. (2022), we also compare SA-MLP with other inference acceleration techniques with GNNs, including vanilla SAGE, quantized SAGE from FP32 to INT8 (QSAGE), SAGE with 50% weights pruned (PSAGE), and inference with neighbor sampling with fan-out 15 (NSSAGE). All GNNs have three layers and 256 hidden units, while GLNN+ has 1024 hidden units for Arxiv and 2048 for Products to achieve optimal performance Zhang et al. (2022). As shown in Table 8, all MLP-like students achieve substantially faster inference than GNN variants. Moreover, SA-MLP is the only method that processes structured inputs explicitly but offers the fastest inference speed without sacrificing performance. Following the original paper of NOSMOG, we set the hidden size to 256 and achieved the best performance. Although it processed the pre-compputed deepwalk embedding, the Setting the hidden Without considering the time cost of preprocessing by DeepWalk, NoSMOG's inference time is significantly less than that of GLNN. The reason is that it uses 256 as the hidden size to achieve optimal performance. Compared to GLNN+ and NOSMOG, the speedup can be attributed to Pytorch's sparse tensor multiplication with structure inputs and the smaller hidden unit size (128 for Arxiv and 64 for Products) employed in SA-MLP.

