# OpenReview forum: "SA-MLP: Distilling Graph Knowledge from GNNs into Structure-Aware MLP"
_TMLR — Accepted by TMLR_

### Review · Reviewer_csBk · 2024-04-09

**Summary Of Contributions:**

In this paper, the authors introduce a strategy to perform knowledge distillation for graph machine learning, which allows to transfer knowledge from GNNs to simple MLPs, which are far faster at the inference stage. While existing methods incorporate the graph structure as deepwalk embedding or ignore it altogether, the authors adopt a simple strategy in which the columns of the adjacency matrix are simply concatenated as features to each node. They also propose a mixup data augmentation strategy. They perform extensive experiments in the transductive and online settings, with ablation studies and many different datasets.

**Audience:**

No

**Claims And Evidence:**

No

**Requested Changes:**

See above, the proposed architecture is specific to one fixed graph, which is exceedingly restrictive.

**Strengths And Weaknesses:**

Strengths:
- an interesting idea
- an extensive set of experiments

Weakness:
- my main concern/confusion is about the architecture itself. Unless I am missing something, by considering the column/lines of the adjacency matrix as features (equation (5), in which I think it is the lines of $A$ that are features and not the columns as indicated by the authors), the trained MLP is specific to one graph. In particular, it is not permutation-equivariant (which is the main characteristic of graph machine-learning models), and cannot be applied to new graphs, graphs with different number of nodes (the dimension of $W_A$ will not match), and so on. For this matter, I do not understand how the online experiment are performed, since the training and testing graphs do not have the same dimension?

This seems very restrictive, and is in fact somewhat of a "forbidden" operation in graph machine-learning. Instead, people have to rely on embeddings such as Deepwalk, which are permutation-equivariant and can be computed/generalized on any graphs.

---

> ### Author Response · Authors · 2024-06-27
> **Reponse to Q1:  Limited to one fixed graph.**
>
> Thank you for your insightful question. Like other methods that distill GNNs into MLPs, our approach focuses on a single online large-scale graph, which is a common scenario in industrial applications (e.g., large-scale online e-commerce recommendation systems). In the large online graph, each node is assigned a unique static ID. As mentioned in the paper, $W_A$ can be viewed as a learnable embedding table for each node, so the issue of permutation equivariance does not arise under the unique static ID of nodes.
>
> Our approach is not suitable for graph learning scenarios where the training and testing sets consist of many different graphs, such as in molecular graphs, which typically contain only a few hundred nodes and have unique structural characteristics. However, in these graphs scenarios,  inference latency is generally not a concern, and thus, there is no need to distill GNNs into faster MLPs. Consequently, such applications fall outside the scope of this paper.
>
> In equation (5), the rows of $A$ represent samples, and the columns of $A$ represent features. In online experiments, following the approach of NOSMOG, inference is performed on new nodes appearing in the graph. These new nodes form connections with existing nodes (e.g., new users on Twitter following recommended accounts). This means the new nodes add rows to $A$, but they share the same feature space (columns of $A$), allowing $W_A$ to support inference on these new nodes. Specifically, during online inference, as described in NOSMOG, we use the connections between new and existing nodes to extract the node structure embeddings from $W_A$ to make predictions for the new nodes. Furthermore, as the number of new nodes increases, we can periodically expand the columns of $A$ and retrain to learn the structure embeddings of the new nodes. This parameter expansion approach is also common in large-scale recommendation systems, making our method flexible to support a growing large-scale online graph.

---

### Review · Reviewer_XdoB · 2024-04-20

**Summary Of Contributions:**

This paper proposes a novel structure-aware MLP as the student model for graph knowledge distillation. The technique includes a structure-mixup technique that can generate virtual distillation samples. Experiments on multiple datasets and different setups verify the validity of the method.

**Audience:**

Yes

**Claims And Evidence:**

No

**Requested Changes:**

1. I don't quite get what the $X_*$ means in equation 7. If it is randomly shuffled among the whole graph, does it mean that features from two structurally irrelevant nodes are likely to be added up in $X_*$? If so, it seems not reasonable.

2. As a contribution of this work, the ablation study of whether to use structure-mixup or not seems not included. Meanwhile, there is no experimental study on how to choose $\gamma$ in equation 7. I only find the experiments for $\alpha$ in Figure 4.

3. I am wondering whether you can use deeper GNN models for the teacher model. What will the student model change accordingly in this case? Can you use Graph Transformer or attention-based GNN in the teacher model?

**Strengths And Weaknesses:**

Strengths:
1. The paper is overall easy to follow and understand.
2. The problem is interesting, and the proposed method is effective in multiple settings.

Weaknesses:
1. The explanation of the structure mixup seems not clear or reasonable enough.
2. The studied models are not large enough.

---

> ### Author Response · Authors · 2024-06-27
> **Response to Q1：Randomly mixup seems not reasonable.**
>
> Thank you for this good question. Our choice to not specifically consider the structural relevance between nodes when performing mixup is driven by the simplicity and generality of the mixup approach. Let's recall that Mixup, when applied in image processing (where images can be seen as a special kind of graph), involves randomly mixing two samples without considering their relationship and semantic similarity[1].
>
> The reason this approach works is that when we mix the features X of samples, we simultaneously mix their labels Y in a consistent manner. This random, unbiased mixing allows the model to learn from a diverse set of mixed samples, thereby enhancing its generalization capability, regardless of the relationship between the original samples.
>
> Additionally, calculating structural similarity and selecting samples based on this similarity for mixup is indeed an interesting direction. However, the computation of structural similarity and the selection of samples would introduce extra overhead. For the sake of efficiency, our current work adopts the simplest form of random mixup. In future work, we plan to explore the impact of incorporating structural similarity into the mixup process.
>
> [1]Zhang H, Cisse M, Dauphin Y N, et al. mixup: Beyond Empirical Risk Minimization[C]. ICLR 2018

---

> > ### Author Response · Authors · 2024-06-27
> > **Response to Q2：Ablation study of structure-mixup and $\gamma$.**
> >
> > Thanks for your question. Both of these experiments are actually included in our paper. The ablation study for structure-mixup can be found in Table 4 and Section 5.3.2 (variants d and e), where we compare the results with and without the SKD component. As shown, the use of SKD consistently provides benefits over standard KD across different scenarios.
> >
> > Regarding $\gamma$ in equation 7, it acts as a mixing coefficient in the mixup process and is dependent on $\alpha$. Specifically, as shown in equation 7, $\gamma$ is sampled from a Beta distribution parameterized by $\alpha$. Therefore, the experiments on $\alpha$ presented in Figure 4 essentially investigate the impact of $\gamma$ as well.

---

> > > ### Author Response · Authors · 2024-06-27
> > > **Reponse to Q3: How about Deeper GNN, Graph Transformer and GAT as teacher.**
> > >
> > > Thank you for this insightful question. We have indeed conducted related experiments, as shown in Figure 3 of the original main text. Overall, SA-MLP as a student consistently outperforms various types of teacher GNNs after distillation.
> > >
> > > Specifically, for deeper GNNs, we used the popular GCNII as the teacher model. As observed, GCNII (represented in yellow) performs better as a teacher compared to other GNNs. However, the distillation results to different MLPs might not be as good as using a shallower GNN like SAGE as the teacher. This could be because the deeper the GNN, the greater the architectural difference between the GNN and the MLP, which might result in a lower generalization capability for the distilled MLP.
> > >
> > > For attention-based GNNs, the GAT model shown in Figure 3 is an attention-based GNN. It can be seen that SA-MLP achieves better distillation performance from attention-based GNNs compared to other MLP students. This is an interesting observation, possibly because SA-MLP has more learnable structural parameters and fitting capabilities.
> > >
> > > Regarding Graph Transformers, due to their quadratic complexity on the number of nodes, they are typically used in small graphs such as molecular property prediction tasks[1]. Training a Graph Transformer like Graphormer on large-scale graphs is challenging, so they are generally not used as teachers in large-scale graph datasets.
> > >
> > > [1] Ying C, Cai T, Luo S, et al. Do transformers really perform badly for graph representation?[C]. NIPS 2021.

---

### Review · Reviewer_L4dK · 2024-06-13

**Summary Of Contributions:**

Due to the inference latency of GNNs in large-graphs, this paper presents a knowledge distillation technique to train a Structure-Aware MLP. The proposed approach is based on generating virtual samples using the structural knowledge from teacher GNNs.

**Audience:**

Yes

**Claims And Evidence:**

No

**Requested Changes:**

Please see weaknesses.

**Strengths And Weaknesses:**

This paper is well-written and easy to follow. Also, the topic of knowledge distillation for GNNs has currently attracted much attention with several recent attempts to address its challenges. Therefore, I believe that the paper is interesting for at least some individuals in TMLR's audience. The proposed method is relatively novel and has shown promising results in different tasks and settings.


There are, however, several concerns with the evaluations and motivations. There is a lack of discussion with several recent methods (to name a few [1, 2]) that also have discussed the importance of inductive setting, showing good results in this setting as well. Not only a discussion of these studies is necessary, but it also is important to compare them with this method and provide evidence as to why we need SA-MLP in the presence of the existing methods.

Currently, there is no theoretical evidence for the performance of SA-MLP and why it can achieve good performance. When talking about large graphs, expressive models usually show better results. Is there any guarantee for the expressive power of SA-MLP? Can it preserve the power of teacher GNNs?

To my understanding, the proposed method uses more memory than existing approaches. This can be an important challenge for large graphs, which is the interest of this paper. It would be beneficial to have a memory comparison with existing methods.


[1] VQGraph: Rethinking Graph Representation Space for Bridging GNNs and MLPs. Yang et al. ICLR 2024.
[2] Structure-aware Semantic Node Identifiers for Learning on Graphs. Leo et al. 2024

---

> ### Author Response · Authors · 2024-06-27
> **Response to Q1: Discussion with [1] and [2]**
>
> Thank you for pointing out these relevant works. We will include and discuss them in our revised manuscript. Specifically, both [1] and [2] employ a two-stage learning approach based on codebooks. They draw inspiration from VQVAE by first pretraining a codebook using the hidden layer representations of GNNs to capture structural information, and then using this pre-trained codebook for downstream tasks. While this method achieves good results, it has some limitations:
>
> 1. **Dependency on the Hidden Layer and Structure of GNN :** These methods require knowledge of the specific GNN architecture to leverage hidden layer information for codebook learning,  which can be limiting in scenarios where only model outputs are accessible, particularly with proprietary large models.
> 2. **2-Stage Pretraining Overhead:** Pretraining on large-scale graphs using GNNs for codebook is also time-consuming and resource-intensive. As noted in the VQGraph study, specific sampling techniques are needed to scale to large graphs.
>
> Our method, SA-MLP, offers several distinct advantages:
>
> 1. **Architecture Independence:** It does not rely on the specifics of GNN architecture, only requiring model output labels for distillation, enhancing versatility.
> 2. **Efficiency:** Our method facilitates end-to-end learning in a single-stage distillation process, optimizing speed and reducing complexity, ideal for industrial settings.
> 3. **Innovative Distillation Strategy:** We introduce a novel mixup-based structure-aware distillation technique that improves performance by effectively leveraging structural information.
>
> These features demonstrate the practicality and necessity of SA-MLP.
>
> [1] Yang et al. VQGraph: Rethinking Graph Representation Space for Bridging GNNs and MLPs[C].  ICLR 2024.
>
> [2] Leo et al. Structure-aware Semantic Node Identifiers for Learning on Graphs[J]. arXiv  2024

---

> > ### Author Response · Authors · 2024-06-27
> > **Response to Q2: Theoretical evidence.**
> >
> > Thank you for the insightful suggestion. Providing theoretical guarantees for the performance retention in cross-model architecture knowledge distillation is indeed challenging. Therefore, this work focuses on proposing a simple yet effective model suitable for large-scale industrial applications. From an experimental perspective, we have compared the performance of distilling various teacher GNNs into SA-MLP on graphs with different properties (homophily, heterophily) and sizes under both transductive and online settings. The extensive experimental results demonstrate that SA-MLP consistently maintains or even surpasses the performance of GNNs.
> >
> > While providing theoretical guarantees is beyond the scope of this paper, it is worth noting that most GNNs operate as a form of Laplacian smoothing mechanism[1]. According to the analysis of GNNs from the perspective of label smoothing[2], the smoothing effect on labels is theoretically bounded by the differences in features between nodes and their neighbors. In future work, we could explore the notion of treating the soft labels provided by teacher GNNs as a form of label smoothing[3]. From this perspective, we can investigate how structural information is retained by SA-MLP and other MLP-based student models through feature and structural embeddings of individual nodes.
> >
> > [1] Nt H, Maehara T. Revisiting graph neural networks: All we have is low-pass filters[J]. arXiv 2019.
> > [2] Wang H, Leskovec J. Combining graph convolutional neural networks and label propagation[J]. ACM Transactions on Information Systems 2021.
> > [3] Yuan L, Tay F E H, Li G, et al. Revisiting knowledge distillation via label smoothing regularization[C]. CVPR 2020.

---

> ### Author Response · Authors · 2024-06-27
> **Response to Q3: Memory usage.**
>
> Thank you for raising this concern. In large-scale industrial recommendation systems, trading space for time is a common practice. For instance, learning a user ID embedding table of size N×dN \times dN×d is a typical approach to store interaction information for efficient inference. This large-scale embedding table storage is well-supported in the industry with mature distributed engineering practices, i.e., parameter servers for distributed training.
>
> As we mentioned in our paper, our approach involves end-to-end learning of a structure embedding for each node. This allows us to store structural information via distillation and support fast inference on large graphs. The embeddings can be stored in a parameter server to reduce memory overhead.
>
> Suppose the Batch size=B, Hidden and structure embedding dimension=d, Average degree of each node=D, Layer=1. We provide the analysis of memory usage, focusing on the differences in how structure embeddings are managed since all models utilize MLP architectures:
>
> - GLNN: Does not require structure embedding or additional storage. Training and inference involve a straightforward MLP, with memory usage of O(Bd).
> - NOSMOG: Requires additional storage for precomputed positional embeddings from DeepWalk: O(Nd). During training and inference, the model processes positional embeddings for a batch of B nodes, resulting in memory usage of O(Bd).
> - SA-MLP: Requires additional storage for all node structure embeddings WA: O(Nd). During training and inference, WA can be stored as an embedding table in parameter servers, and only the required parameters for each batch are retrieved. Based on the batch's lines of adjacency matrix A, the model extracts structure embeddings of B nodes' D neighbors from the weight matrix WA: O(BDd).
>
> [1] Li M, Zhou L, Yang Z, et al. Parameter server for distributed machine learning[C]. Big learning NIPS workshop. 2013.

---

### Decision · Action_Editor_niTY · 2024-07-30

**Recommendation:** Accept as is

**Comment:**

The paper is very borderline, but there are no strong specific cases of claims not matching evidence, so I'd recommend acceptance by tmlr criteria.

**Audience:**

Yes, though I agree with the reviewers that the strong assumptions make the applicability narrow.

**Claims And Evidence:**

Two of the three reviewers think that the evidence matches the claims. The final reviewer remains unhappy about the mix-up operation and thinks it may fail sometimes, but I don't think this is a reason to reject.